# Screening of EWI-2-Derived Peptides for Targeting Tetraspanin CD81 and Their Effect on Cancer Cell Migration

**DOI:** 10.3390/biom13030510

**Published:** 2023-03-10

**Authors:** Thanawat Suwatthanarak, Kei Usuba, Kotomi Kuroha, Masayoshi Tanaka, Mina Okochi

**Affiliations:** 1Department of Chemical Science and Engineering, Tokyo Institute of Technology, 2-12-1-S1-24 O-okayama, Meguro-ku, Tokyo 152-8552, Japan; 2Siriraj Cancer Center, Faculty of Medicine Siriraj Hospital, Mahidol University, 2 Wanglang Road, Bangkok Noi, Bangkok 10700, Thailand; 3Department of Surgery, Faculty of Medicine Siriraj Hospital, Mahidol University, 2 Wanglang Road, Bangkok Noi, Bangkok 10700, Thailand; 4Department of Chemical Science and Engineering, Tokyo Institute of Technology, 4259 Nagatsuta-cho, Midori-ku, Yokohama-shi 226-8501, Kanagawa, Japan

**Keywords:** peptide, CD81, tetraspanins, cancer, migration

## Abstract

CD81, a transmembrane protein belonging to the tetraspanin family, has recently been suggested as a therapeutic target for cancers. Here, we screened peptides that bind to the tetraspanin CD81 protein, and evaluated their inhibitory activity in cancer cell migration. To screen for CD81-binding peptides (CD81-BP), a peptide array membrane was prepared from the amino acid sequence of the EWI-2 protein, a major partner of CD81, before binding to fluorescently labeled CD81. As a result, four candidate CD81-BPs were identified and characterized. In particular, the CFMKRLRK peptide (called P152 in this study) was found to be the best candidate that preferentially binds to the extracellular loop of CD81, with an estimated dissociation constant of 0.91 µM. Since CD81 was reported to promote cancer cell migration, an initial step in metastasis, the Boyden chamber assay, was next performed to assess the effect of CD81-BP candidates on the migration of MDA-MB-231 human breast cancer cells. Interestingly, our result indicated that P152 could suppress MDA-MB-231 cell migration at the level comparable to that of an anti-human CD81 antibody (5A6). Thus, we propose these CD81-BPs with the anti-migration property against cancer cells for the development of novel therapeutic strategies.

## 1. Introduction

In recent years, much research has been focused on tetraspanins, a family of membrane proteins found in multicellular eukaryotes [1,2,3,4,5]. Tetraspanins share a similar topology that includes four transmembrane domains, a small extracellular loop, and a larger extracellular loop (LEL) [4]. As important membrane organizers, tetraspanins have been reported to play a key role in a broad range of biological functions, such as cell fusion, adhesion, motility invasion, infection, and signal transduction, through the formation of tetraspanin webs, multi-protein complexes among tetraspanins and their partner proteins [5].

Among all human tetraspanins, the CD81 protein has been involved in some pathological conditions, particularly viral infection and cancer metastasis [6,7,8,9,10,11,12]. For example, the CD81 protein was recognized as a cell-surface receptor for the hepatitis C virus (HCV) through the binding interaction between the HCV envelope protein E2 and CD81 LEL [6,7,8]. The expression level of CD81 was found to be closely correlated with the vulnerability of host cells to HCV infection [9]. Interestingly, in the field of cancer, CD81 was expressed in many malignancies, including breast cancer, lung cancer, prostate cancer, melanoma, lymphoma, and brain tumor, and its overexpression was correlated with a poor prognosis [10]. Additionally, CD81 was abundantly expressed in breast cancer tissues compared to normal tissues [11]. The increase in CD81 expression was significantly associated with clinical stage lymph node metastasis, and overall survival rate in patients with breast cancer [11]. CD81 also contributed to motility, invasion, and metastasis of melanoma, by inducing the expression of membrane type 1 matrix metalloproteinase in the Akt-dependent Sp1 activation signaling pathway [12]. Taken together, tetraspanin CD81 has the potential to be therapeutically targeted.

Targeting CD81 with a monoclonal CD81 antibody showed promise for reducing HCV infection in cynomolgus monkeys [13]. In addition, anti-human CD81 antibody (5A6) could effectively inhibit in vitro migration and invasion, as well as in vivo metastasis of MDA-MB-231 human breast cancer cells [14]. The therapeutic utilization of antibodies remains subject to some major limitations, including large molecule size, susceptible degradation, side effects, and possible resistance [2,15]. Compared to other biomolecule probes, peptide (i.e., a short chain of amino acids) has a small molecule size, high stability, and relatively low cytotoxicity, suggesting the potential use of peptides in the targeting of CD81 [16,17,18].

Previously, we used peptide array technology to screen peptides for targeting CD9, the other tetraspanin, from its EWI-2 partner [18,19,20]. Nevertheless, the EWI-2 protein is also a major partner for CD81 [21]. Therefore, in this study, our aim was to screen CD81-binding peptides (CD81-BP) from the amino acid sequence of the EWI-2 protein and to assess their impact on cancer cell migration. As a result of the binding assay between the EWI-2-derived peptide array and fluorescently labeled CD81, we were able to identify four candidate CD81-BPs from the EWI-2 sequence. After characterization of CD81-BP candidates, their inhibition effect on MDA-MB-231 cancer cell migration was later demonstrated in the Boyden chamber assay. We believe that these eight-mer CD81-BPs would be alternative probes for therapeutically targeting the CD81 membrane protein.

## 2. Materials and Methods

### 2.1. Chemicals and Materials

All chemicals and materials used in this study were of analytical and research grade. The aqueous solution was prepared using ultrapure water (Milli-Q; Merck Millipore, Burlington, MA, USA). The cellulose membrane (Grade 542) was purchased from Whatman, Buckinghamsh, UK. N-Fluorecenyl-9-methoxycarbonyl (Fmoc)-protected L-amino acids, 20% piperidine in N, N’-dimethylformamide (DMF), DMF, hydroxybenzotriazole (HOBt), diisopropylcarbodiimide (DIPCI), triisopropylsilane (TIPS), O-(benzotriazole-1-yl)-N, N, N’, N’-tetramethyluronium hexafluorophosphate (HBTU), N, N-diisopropylethylamine (DIPEA), m-cresol and trifluoroacetic acid (TFA) were obtained from Watanabe Chemical, Osaka, Japan. Hydrochloric acid (HCl), 2-amino-2hydroxymethyl-1,3-propanediol (Tris), 1-methyl-2-pyrrolidone (NMP), 99.5% ethanol (EtOH), acetic anhydride, dichloromethane (DCM), diethyl ether, acetonitrile and bovine serum albumin (BSA) were obtained from Fujifilm Wako Pure Chemical Industries, Ltd., Osaka, Japan. 1,2-Ethandithiol (EDT) and thioanisole were purchased from Tokyo Chemical Industry Co., Ltd., Tokyo, Japan. Acetonitrile was obtained from Sigma-Aldrich, Burlington, MA, USA. Recombinant CD81 human protein (Cat. No. 14244-HNCH, two additional amino acids (Gly and Pro) to the N-terminus of LEL (Phe113-Lys201) of CD81) was purchased from Sino Biological, China. The Alexa Fluor ™ 488 protein labeling kit was purchased from Thermo Fisher Scientific, Waltham, MA, USA. Dulbecco’s modified eagle medium (D-MEM) and fetal bovine serum (FBS) were obtained from Life Technologies, Carlsbad, CA, USA. The CD81 monoclonal antibody (5A6, Cat. No. sc-23962) was purchased from Santa Cruz Biotechnology, Santa Cruz, CA, USA.

### 2.2. Preparation of the EWI-2 Peptide Array

The peptide array was prepared similarly to the method described in previous reports [18,19,20]. A peptide library of eight-residue peptides was obtained by overlapping four residues from Ig3 to the C-terminus of human EWI-2 (accession number: Q969P0) (Appendix A). Next, the EWI-2 peptide array was synthesized on a Fmoc-β-Alanine-activated cellulose membrane using Fmoc-based chemistry. The synthesis cycle was initiated by deprotecting the N-terminal Fmoc group with 20% piperidine in DMF and then washing it with DMF and EtOH, respectively. The Fmoc amino acid in NMP was next activated by HOBt and DIPCI, before being coupled to the deprotected N-terminus on the membrane within a peptide auto-spotter (MultiPep RSi; Intavis, Tübingen, Germany). Upon completion of the double coupling, the uncoupled amino groups were capped with 4% acetic anhydride in DMF, followed by membrane washing with DMF and EtOH, respectively, at the end of the synthesis cycle. After elongation of the final amino acid, the Fmoc and side chain protectants were removed with 20% piperidine in DMF, and a mixture of ultrapure water, TIPS, and TFA (2:3:95, *v/v/v*), respectively. The membrane was finally washed with DCM, DMF, ethanol, and phosphate-buffered saline (PBS, pH 7.4), respectively.

### 2.3. Preparation of Fluorescently Labeled CD81 Protein

The Alexa Fluor™ 488 protein labeling kit (Thermo Fisher Scientific, Waltham, MA, USA) was applied to fluorescently label the CD81 protein. According to the user guide, 84 mg of sodium bicarbonate was first dissolved in 1 mL of Milli-Q to prepare 1 M sodium bicarbonate solution. The CD81 solution was then prepared at 0.2 mg/mL by dissolving 100 µg of CD81 in 500 µL of 0.1 M sodium bicarbonate buffer (pH 8.3). The obtained CD81 solution was then applied to a vial containing Alexa Fluor™ 488 fluorescent dye with a succinimidyl ester group, followed by incubation on a magnetic stirrer at room temperature in the dark for 3 h. The excess dye was purified using the Slide-A-Lyzer™ dialysis cassette (3.5K MWCO, 0.5 mL, Thermo Fisher Scientific, USA) and 100 mM Tris–HCl buffer. The concentration of the labeled CD81 protein solution was finally quantified using a Bradford protein assay kit (Takara Bio Inc., Shiga, Japan).

### 2.4. Binding Assay between EWI-2 Peptide Array and Fluorescently Labeled CD81 Protein

First, to reduce nonspecific interactions, the peptide array was blocked by 3% BSA in Tris–HCl buffer at room temperature for 15 min, followed by membrane washing with Tris–HCl buffer. The peptide array was then immersed in 10 mL of 500 nM fluorescently labeled CD81 solution at room temperature for 3 h, followed by membrane washing with Tris–HCl buffer. Finally, the membrane was fluorescence-scanned and imaged by a biomolecular imager (Typhoon FLA 9500; GE Healthcare, Hatfield, UK), before the spot intensity of each peptide was measured using GE Healthcare’s ImageQuant TL 8.1 software.

### 2.5. Characterizations of Candidate Peptides

For evaluation of the dissociation constant (K_D_), the peptide array containing candidate CD81-BPs and a control AAAA peptide was prepared, as described in Section 2.2. After blocking the peptide array with 3% BSA, the peptide array was immersed in a solution of Alexa488-labeled CD81 protein (1 µM, 750 nM, 500 nM, 250 nM, 125 nM or 50 nM) at room temperature for 3 h. The peptide array was then washed with Tris–HCl buffer before scanning with Typhoon FLA 9500 imager (GE Healthcare, UK). The spot intensity of each peptide was then quantified by the ImageQuant software (GE Healthcare, UK). In the final step, the K_D_ value of each candidate peptide for the CD81 protein was determined using Origin, data analysis, and graphing software (OriginLab Corporation, Northampton, MA, USA). The K_D_ was estimated by the Hill correlation, according to the following equation:y = y_max_(x^n^)/(k^n^ + x^n^),
where y is the intensity, y_max_ is maximum intensity, x is concentration, n is the Hill coefficient and k^n^ is K_D._

To demonstrate the binding site based on protein structure and peptide sequence, the complex structure of each CD81-BP candidate and CD81 was simulated by MDockPeP, a protein–peptide docking server [22]. All CD81-BP candidates were also subjected to a hierarchical algorithm-based web server (called HPEPDOCK) for blind peptide–protein docking, to assess the selective affinity of each CD81-BP candidate to CD81 and CD9 using a docking energy score (a mean of the top ten models) (http://huanglab.phys.hust.edu.cn/hpepdock/, accessed on 13 September 2022) [23]. Furthermore, the HPEPDOCK server was also used to assess the affinities of P152 to other cancer cell surface proteins, including EpCAM and CD44.

### 2.6. Preparation of Candidate Peptide Powders

The peptide powder was prepared similarly to the method in an earlier report [20]. Multi-column peptide synthesizer (ResPep SLi; Intavis, Germany) and Fmoc-Rink-Amide-(aminomethyl)-Resin were used. In the first step for each amino acid extension, the N-terminal Fmoc group was deprotected with 20% piperidine in DMF, and then the resin was washed with DMF. Then, Fmoc amino acid in DMF was activated by DIPEA and HBTU, followed by coupling to the N-terminal. Acetic anhydride (5%) in DMF was applied to cover the unreacted amino groups prior to washing the resin with DMF. After the final extension and the Fmoc deprotection by 20% piperidine in DMF, the resin was soaked in a cleavage cocktail (TFA:Milli-Q:thioanisole:m-cresol:EDT:TIPS, 82.5:5:5:5:2.5:1) and incubated for 3 h. The peptide was separated from the resin using a syringe filter and precipitated by ice-cooled diethyl ether. The peptide was then pelleted by 5000 rpm centrifugation at 0 °C for 10 min, followed by washing with ice-cooled diethyl ether. The peptide pellet was finally dissolved in 30% acetonitrile and lyophilized overnight. The obtained peptide powders were purified by a high-performance liquid chromatography (HPLC) system (Shimadzu Corporation, Japan), using an ODS-80TS column (Tosoh Corporation, Japan) and an acetonitrile–water gradient (10 to 90% acetonitrile). The peptide detection was performed with absorption at 214 nm. The molecular weight of the purified peptide powder was confirmed by the mass spectrometry (MS) technique (Appendix A). Subsequently, 90% or more purity of the identified peptide was ensured by an HPLC system equipped with an ODS-100Z column (Tosoh Corporation, Tokyo, Japan) (Appendix A). The prepared peptide powder was stored at 4 °C until use.

### 2.7. Boyden Chamber Assay

A Boyden chamber assay was performed using the CytoSelect™ 96-well cell migration assay kit (8 µm, Fluorometric Format, Cat. No. CBA-106; Cell Biolabs, Inc., San Diego, CA, USA). Human breast cancer MDA-MB-231 cells (HTB-26TM; American Type Culture Collection, Manassas, VA, USA) were suspended in serum-free D-MEM medium (1.85 × 10^5^ cells/mL) containing 100 nM CD81-BP, AAAA or CD81 antibody (5A6) at 37 °C and 5% CO_2_ for 1 h. Then, within the Boyden chamber, 100 µL of the serum-free cell suspension was added to the upper chamber, while 150 µL of 10% FBS-containing medium was placed in the lower chamber. After incubation at 37 °C and 5% CO_2_ for 24 h, the cells that migrated to the lower chamber were detached from the membrane by cell detachment buffer, lysed by lysis buffer, and stained with CyQuant^®^ GR dye. The fluorescence intensity of the cell lysate was analyzed at 480/520 nm by a microplate reader (Biomedical Solutions, Inc., Stafford, TX, USA).

### 2.8. Statistical Analysis

The experiment was performed with at least three replicates. The *t*-test was performed using Prism 9 (GraphPad Software, Inc., La Jolla, CA, USA) or Microsoft Excel (Microsoft, WA, USA) to analyze the statistical significance difference. The statistical significance was set at *p*-value < 0.05.

## 3. Results and Discussion

### 3.1. Screening and Characterization of CD81-BP Candidates

EWI-2, or immunoglobulin superfamily member 8 protein, has highly and specifically been associated with CD9 and CD81 [10,21], suggesting that it is a strong candidate for screening CD81-BPs. The EWI-2 peptide array contained eight-residue peptides derived from the EWI-2 sequence by a four-residue frameshift from the Ig3 domain to the C-terminus, where it was necessary for associating with the LEL of CD81 tetraspanin (Appendix A) [24,25]. Figure 1A shows the peptide array after binding assay with fluorescently labeled CD81 protein. The fluorescence spots could be clearly observed, and their intensity could be quantified by image-analyzing software (Figure 1A). The fluorescent dye solution (at the same concentration as the labeled protein solution) was also used in the binding assay as a blank control (Figure 1B). Compared to the peptide array bound to the labeled protein (Figure 1A), the peptide array bound to only fluorescent dye (Figure 1B) had no distinct spot intensity, implying no or relatively low nonspecific binding from the dye.

Figure 2 illustrates the blank-subtracted spot intensity of each peptide, which represented the interaction between each peptide and the CD81 protein. From this result, four peptides (P97, P132, P152, and P153) with a fluorescence intensity greater than the average (AVG) and two standard deviations (SD) were selected as candidate CD81-BPs (Figure 1A, Figure 2 and Table 1). Polar and electrostatic properties were considered important for the interaction of these candidates with the CD81 protein since all candidates had a negative grand average of hydropathy (GRAVY) value (hydrophilic), an isoelectric point greater than seven, and a positive charge (Table 1). However, the affinity to CD81 of candidate peptides was not proportional to the isoelectric point, charge, or GRAVY values (Figure 2 and Table 1). The four candidate sequences contained the Arg-Leu-Arg (RLR) motif (Table 1), which is usually found in functionally important domains of proteins for binding to anionic phosphate and/or carboxylate groups [26,27]. Therefore, these candidates were potentially derived from the functional sites of the EWI-2 protein. The presence of the RLR motif was also found in candidates for CD9-binding peptides from EWI-2 previously reported [20]. Because the EWI-2 protein is a key partner for both CD81 and CD9 tetraspanins, some previously reported peptides for targeting CD9 were also listed as CD81-BP candidates in this study [20]. Particularly, the RSHRLRLH peptide (P132) showed a good affinity toward CD9 in our previous studies [18,20]. Thus far, we were able to screen these four CD81-BP candidates from the EWI-2 protein for the following characterizations. The peptide arrays containing all candidates were then applied in the binding assay with fluorescently labeled CD81 at different concentrations, to estimate the K_D_ value of each CD81-BP candidate (Appendix A). Appendix A shows the negative control (4A or AAAA peptide)-subtracted spot intensities of all candidates against CD81 concentrations. According to the Hill equation, the estimated K_D_ value of P97, P132, P152, and P153 was 12.6, 1.63, 0.91, and 1.06 µM, respectively (Table 1). This suggested that P152 had the highest binding affinity to the CD81 protein among all candidates.

To illustrate the binding site of CD81-BP candidates on the CD81 protein, a server named MDockPeP was utilized for predicting protein–peptide complex structures [22]. The protein–peptide complex structure was predicted through the protein structure and peptide sequence [22]. All candidates were found to be probably bound to the proximal membrane LEL domain of the CD81 protein (Appendix A). These results were consistent with the protein fragment used in the binding assay, which was only the LEL domain of the CD81 protein. Thus, these candidate peptides were likely to bind to the extracellular LEL, the major association domain of CD81.

Yang et al. have recently reported that HPEPDOCK, a hierarchical algorithm–based web server for peptide–protein docking, provided a strong linear correlation between HPEPDOCK docking energy scores and the in vitro inhibitory activities of monoamine oxidase A peptides docked to the enzyme [23,28]. Their finding was evidenced by using HPEPDOCK’s docking energy scores, in order to evaluate bioactive peptide affinities [28]. Here, the selective binding of CD81-BP candidates to CD81 and CD9 was studied in silico by the HPEPDOCK server [23]. As shown in Figure 3, all candidates had a negative docking energy score for both CD81 and CD9. Although all CD81-BP candidates had the potential to bind to CD9, P152 had a lower docking energy score to CD81 than to CD9, indicating the preference affinity of P152 to CD81 instead of CD9 (Figure 3). These protein–peptide docking results also supported the preference of P132 to CD9 (K_D_ of P132 to CD9 = 0.46 µM estimated in the previous study [20]), rather than CD81 (K_D_ of P132 to CD81 = 1.63 µM estimated in this study). These experimental and computational results suggested the specificity of P132 to CD9 and P152 to CD81. Additionally, P152 showed a preference for CD81 compared to EpCAM and CD44, which are two common cancer cell surface markers (Appendix A). Three P152-scrambled peptides also had a higher docking energy score for CD81 than the original P152, implying the loss of CD81-binding activity in the scrambled peptides (Appendix A). These data indicate that the P152 sequence was specific for CD81 binding, rather than amino acid composition. Collectively, P152 (CFMKRLRK) was a promising peptide from the EWI-2 sequence that preferentially bound to CD81 LEL, with an estimated K_D_ of 0.91 µM.

### 3.2. Effect of CD81-BP Candidates on Cancer Cell Migration

The impact of all CD81-BP candidates on cancer cell migration through the porous membrane was last observed in the Boyden chamber assay. Herein, the MDA-MB-231 cell line was used because this is a triple-negative human breast cancer cell line, with a more aggressive and metastatic behavior compared to other breast cancer cell lines, and shows the high expression of CD81 [14]. MDA-MB-231 cancer cells were incubated with a candidate CD81-BP (100 nM) in serum-free medium before applying them to the upper chamber. The peptide concentration of 100 nM was chosen because a CD9-targeting peptide, screened from EWI-2 in our previous study, could significantly inhibit cancer cell migration in this condition [18]. A normal FBS-containing medium was placed in the lower chamber. The cells were allowed to migrate through a porous membrane for 24 h, before the cells that migrated to the opposite side of the membrane were collected, lysed, and quantified by a fluorescent detection method. Figure 4 shows the fluorescence intensity quantified from the migrated cells. Compared to non-treatment (−), P132, P152, and P153 could significantly reduce the migration of MDA-MB-231 cells, while no distinct inhibition was observed in P97 under this condition (Figure 4). Possibly, it was due to the relatively high K_D_ of P97 that caused no inhibitory effect at the concentration of 100 nM (Figure 4). Surprisingly, the inhibitory abilities of P132, P152, and P153 were comparable or competitive to an anti-CD81 monoclonal human antibody (5A6) [14]. In particular, P152 had the highest potency to suppress MDA-MB-231 cell migration (Figure 4). P132, previously described as a CD9-binding peptide [18,20], also showed anti-migration activity against MDA-MB-231 cancer cells because of the high expression of CD9 in this cell line. Nevertheless, the best CD81-BP candidate or P152 was more effective in inhibiting MDA-MB-231-cell migration than P132 (Figure 4), possibly due to the higher association of EWI-2 to CD81 than CD9 [21]. Furthermore, compared to its two scrambled peptides (P152-1 (FRKCKMRL) and P152-2 (KKRCRFLM)), only P152 origin showed statistical significance (*p*-value < 0.05) in the anti-migration activity (Figure 5, Appendix A). The scrambled P152-1 showed the impact on MDA-MB-23 cell migration as seemingly as the original P152, possibly because both contained the RL nest, a phosphate- and carboxylate-binding site that presented in the functionally important region of EWI-2 protein (Figure 5) [27,28]. As anti-human CD81 antibody (5A6) was previously reported to reduce tumor growth and metastasis in vivo by inhibiting cancer cell migration and invasion [14], P152, with comparable anti-migratory ability to 5A6, would be one of the promising biomolecules for cancer therapy. According to our previously screened CD9-binding peptides, the observation by high-resolution confocal microscopy suggested that the peptides directly bound to CD9 expressed on the cell surface [29]. Thus, we hypothesize that P152 in this study has the same potential to directly bind to CD81 expressed on the cell surface and then inhibit cancer cell migration. Due to the limited data here, the affinity, specificity, and inhibitory activity of P152 should be extensively studied by various investigations, such as co-immunoprecipitation, flow cytometry, and confocal microscopy assays, in the future. Further P152-based modification and functionalization, such as sequence optimization and structure modification, may enhance its anti-migration potency [30]. P152 suppressing cancer cell migration may reduce the metastasis ability of cancer cells [29]. Because tetraspanin CD81 protein is one of the biomarkers for extracellular vesicles, P152 may also be applicable for targeting or collecting extracellular vesicles.

## 4. Conclusions

This is the first attempt to identify CD81-BP from the EWI-2 protein, a major partner of CD81, using peptide array technology. As a result, four candidates (P97, P132, P152, and P153) were identified from the amino acid sequence of EWI-2. The binding site, K_D,_ and selective binding of each CD81-BP candidate were later characterized. Among all candidates, the P152, or CFMKRLRK, peptide had the highest preferential affinity to CD81, with an estimated K_D_ of 0.91 µM using a peptide array. Furthermore, P152 was able to effectively reduce the migration of MDA-MB-231 cancer cells. The affinity and specificity of CD81-BPs should be clarified in future studies. Moreover, the inhibitory mechanism of CD81-BPs for cancer cell migration and *in vivo* assays should be considered. We expect that CD81-BPs proposed here will be useful to target tetraspanin CD81 in a wide range of biological and medical applications.

## Figures and Tables

**Figure 1 biomolecules-13-00510-f001:**
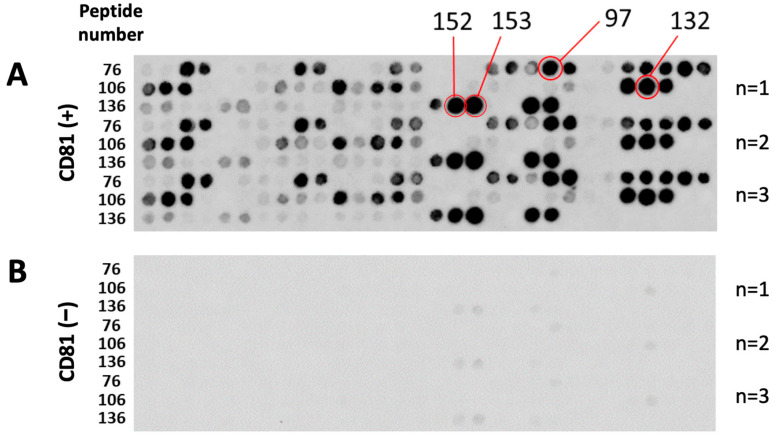
Fluorescent images of EWI-2 peptide array after binding assay with Alexa Fluor^®^ 488−labeled CD81 protein. (**A**) EWI-2 peptide array bound to Alexa Fluor^®^ 488−labeled CD81 protein (CD81 (+)). (**B**) EWI-2 peptide array bound to Alexa Fluor^®^ 488 dye (CD81 (−), blank). The red circles indicate CD81−BP candidate spots.

**Figure 2 biomolecules-13-00510-f002:**
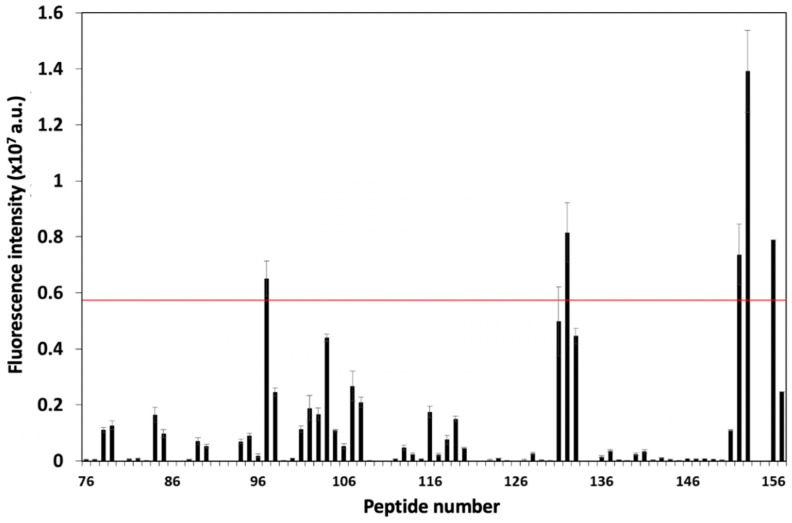
Blank subtracted fluorescence intensity profile of each peptide after binding assay with Alexa Fluor^®^ 488−labeled CD81 protein. Error bars show SD (n = 3). The red line indicates AVG + 2SD. Peptide numbers 76 to 153 were EWI-2−derived peptides. Peptide numbers 154 to 157 were control peptides, AAAA, DDDD, RRRR and KKKK, respectively.

**Figure 3 biomolecules-13-00510-f003:**
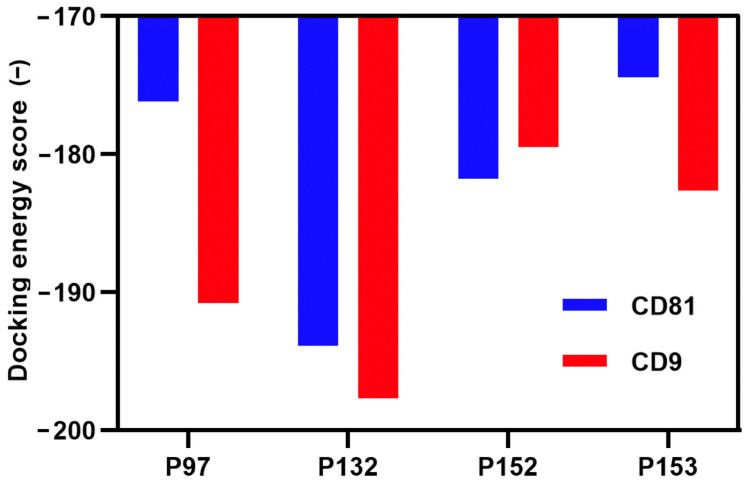
Docking energy scores of CD81-BP candidates for CD81 and CD9 proteins were assessed using a hierarchical algorithm−based web server, HPEPDOCK, for blind peptide–protein docking [23].

**Figure 4 biomolecules-13-00510-f004:**
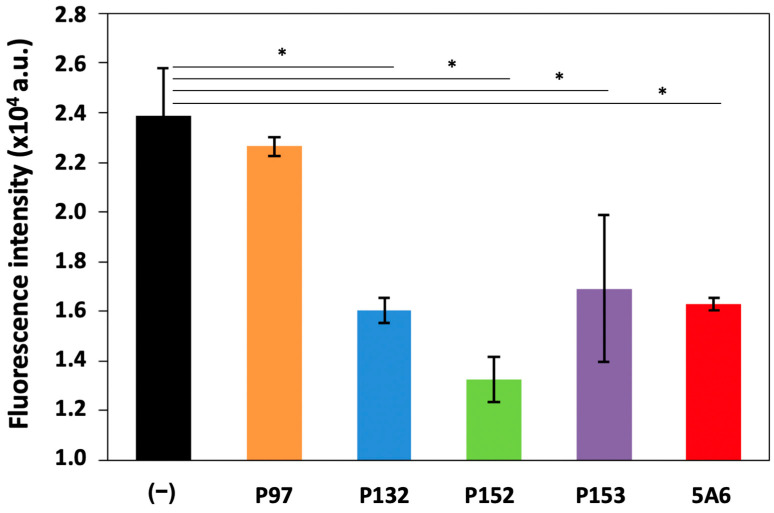
Fluorescence intensity derived from MDA-MB-231 human breast cancer cells that migrated to the opposite side of the membrane in the Boyden chamber assay. MDA-MB-231 cells were treated with a candidate peptide (P97, P132, P152, or P153) or an anti-human CD81 antibody (5A6), at a concentration of 100 nM. (−), no treatment. Error bars show SD (n = 3). *, *p*-value < 0.05, *t*-test.

**Figure 5 biomolecules-13-00510-f005:**
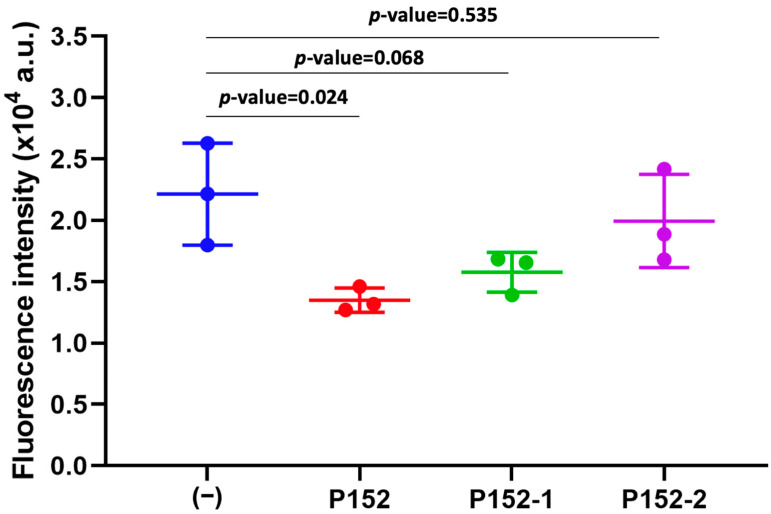
Comparison of the anti-migration activity of P152 to its scrambled sequences. MDA-MB-231 cells were treated with P152 (CFMKRLRK), P152-1 (FRKCKMRL) or P152-2 (KKRCRFLM), at a concentration of 100 nM in the Boyden chamber assay. (−), no treatment. Error bars show SD (n = 3). *p*-value was calculated by a *t*-test.

**Table 1 biomolecules-13-00510-t001:** A list and characteristics of CD81-BP candidates from EWI-2.

Peptide ^a^	Sequence	pI ^b^	Charge ^c^	GRAVY ^d^	Estimated K_D_ (µM) ^e^
P153	FMKRLRKR	12.3	+4.5	−1.6	12.6
P132	RSHRLRLH	12.3	+2.6	−1.6	1.63
P152	CFMKRLRK	11.0	+3.5	−0.72	0.91
P97	ASRTYRLR	11.7	+2.6	−1.3	1.06

^a^ Peptide with a spot fluorescence intensity greater than AVG + 2SD. ^b,c^ Isoelectric point (pI) and charge were obtained from Prot pi, Peptide Tool (https://www.protpi.ch/Calculator/PeptideTool, accessed on 20 September 2022). ^d^ GRAVY, or grand average of hydropathy value, was obtained from GRAVY calculator (http://www.gravy-calculator.de, accessed on 20 September 2022). ^e^ K_D_ was estimated by the Hill equation correlation.

## Data Availability

Additional supporting information can be requested from the corresponding author.

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
