# Peer review of "Screening of EWI-2-Derived Peptides for Targeting Tetraspanin CD81 and Their Effect on Cancer Cell Migration"

_biomolecules, 2023, doi:10.3390/biom13030510_

Round 1
Reviewer 1 Report (Previous Reviewer 2)
None
Author Response
The authors would like to appreciate reviewer 1 for reviewing our manuscript. We have done the minor spell check as you suggested.
Reviewer 2 Report (New Reviewer)
The paper by Suwatthanarak T et al entitled “Screening of EWI-2 derived peptides for targeting tetraspanin CD81 and their effect on cancer cell migration” follows up a similar paper from the group on tetraspanin CD9. This paper entitled “Inhibition of cancer-cell migration by tetraspanin CD9-binding peptide” was published in Chemical Communications in 2021. The rationale of these 2 papers is that tetraspanins CD9 and CD81 both positively regulate cell migration; are overexpressed in some cancers and interact with the same binding partner EWI-2. The authors thus screened peptides able to interfere with the binding of EWI-2 to the tetraspanins in order to prevent cancer cell migration.
My main concern is about specificity of the peptides.
For example, in Figure 1, the specificity of the peptides_said to target CD81_ is controlled only against the alexa fluor dye (Figure 1, panel B) but how much specific these peptides will be against alexa-fluor labelled CD9? The authors should include this control as a new panel C.
Figure 4: The authors control the effect of their peptides on cell migration using an anti-CD81 antibody (5A6). They should also include the CD9 binding peptide described in Chemical Communications as an additional control and maybe test if the anti-CD81 and anti-CD9 peptides could potentiate.
Co-immunoprecipitations experiments showing that the selected peptides indeed reduced the binding of EWI proteins to CD81 will be appreciate.
Author Response
The authors would like to highly appreciate reviewer 2 for reviewing our manuscript as well as providing meaningful comments/suggestions. Based on your comments/suggestions, we have carefully addressed your concerns, especially regarding the peptide specificity, and improved the manuscript as follows.
Point 1:
We would like to apologize for lacking clear descriptions about the specificity of these CD81-binding peptide (CD81-BP) candidates. Therefore, we first would like to address the specificity of these CD81-BP candidates compared to our previous studies. As reviewer 2 mentioned, CD9-binding peptides (CD9-BP) were previously screened and utilized in our previous reports [Ref. no. 18 and Ref. no. 20]. Although the rationale and protocol of this study were similar to an earlier report [Ref. no. 20], most of the CD9-BP candidates in the Ref. no. 20 were not included in the list of CD81-BP candidates in this study. Since EWI-2 protein is a major partner for both CD9 and CD81 proteins, some previously-reported CD9-BP candidates were listed as CD81-BP candidates. Interestingly, RSHRLRLH peptide called P132 in this study was considered as the best CD9-BP in our previous studies [Ref. no. 18, Ref. no. 20]. Nevertheless, the estimated dissociation constant (KD) to CD9 of P132 was 0.46 µM while and the estimated KD to CD81 of P132 was 1.63 µM, suggesting the preference affinity of P132 to CD9 rather than CD81 [Ref. no. 20 and this study]. Moreover, the best CD81-BP candidate or P152 in this study had the lower KD (0.91 µM) to CD81 than P132 (1.63 µM). These results were also consistent with the peptide-protein docking (Figure 3), which showed the preferences of P132 to CD9 and P152 to CD81, respectively. Taken together, the previous and current studies differently provided the best peptide for each target, P132 as the best CD9-BP and P152 as the best CD81-BP.
In response to reviewer 2’s comment, we have clarified the descriptions about the peptide specificity in the revised manuscript as follows.
Lines 237-240: Because EWI-2 protein is a key partner for both CD81 and CD9 tetraspanins, some previously-reported peptides for targeting CD9 were also listed as CD81-BP candidates in this study [20]. Particularly, RSHRLRLH peptide (P132) showed a good affinity toward CD9-BP in our previous studies [18,20].
Lines 279-283: These protein-peptide docking results also supported the preference of P132 to CD9 (KD of P132 to CD9 = 0.46 µM estimated in the previous study [20]) rather than CD81 (KD of P132 to CD81 = 1.63 µM estimated in this study). These experimental and computational results suggested the specificity of P132 to CD9 and P152 to CD81, respectively.
Point 2:
We also used a CD9-binding peptide (RSHRLRLH, called P132 in this study) described in Chemical Communications [Ref. no. 18] in Figure 4. As shown in Figure 4, P132 also showed the anti-migration activity against MDA-MB-231 cancer cells because of the high expression of CD9 in this cell line. However, the best CD81-binding peptide or P152 was more effective to inhibit MDA-MB-231-cell migration than P132 (Figure 4), possibly, due to the higher association of EWI-2 to CD81 than CD9 [Ref. no. 21].
In response to the reviewer 2’s comment, we have added the descriptions in the revised manuscript as follows.
Lines 314-319: P132, previously described as a CD9-binding peptide [18,20], also showed the anti-migration activity against MDA-MB-231 cancer cells because of the high expression of CD9 in this cell line. Nevertheless, the best CD81-BP candidate or P152 was more effective to inhibit MDA-MB-231-cell migration than P132 (Figure 4), possibly, due to the higher association of EWI-2 to CD81 than CD9 [21].
Point 3:
Once again, thank you very much for your suggestion. We totally agree with this suggestion. Actually, we performed the co-immunoprecipitation assays previously. However, we faced the difficulty of recovering CD81, which is a small integral membrane protein. Although we cannot add the co-immunoprecipitation result in the revised manuscript at this time, we have additionally addressed this point as our limitation and also included this point in future investigations as follows.
Lines 332-335: Due to the limited data here, the affinity, specificity and inhibitory activity of P152 should be extensively studied by various investigations, such as co-immunoprecipitation, flow cytometry, and confocal microscopy assays, in the future.

Reviewer 3 Report (New Reviewer)
The manuscript entitled "Screening of EWI-2-derived peptides for targeting tetraspanin CD81 and their effect on cancer cell migration” by Suwatthanarak T et al. showed the identification of CD81- binding peptides, derived from EWI-2 amino acid sequence, by using a peptide array membrane method. Docking studies indicated that among four peptide candidates with highest binding ability to fluorescent CD81 protein on membrane, the P152 peptide possess highest binding affinity for CD81. Additionally, this peptide was able to slightly reduce the migration ability of triple negative breast cancer cells MDA-MB-231 in vitro. These results suggest the promising use of P152 as peptide targeting CD81 protein.
Manuscript is well‐written and concise. The topic is of interest for readers being peptide ligands a promising therapeutics tool in different preclinical and clinical fields. To improve the presentation of manuscript minor changes need to be added.
I suggest that this paper should be accepted for publication in Biomolecules journal with minor revisions.
- Line 63-64: The sentence " Previously, we used peptide array technology to screen peptides for targeting CD9, the other tetraspanin, from its EWI-2 partner " refers to reference 18. Please correct and improve description of references 19-20.
- Line 84: please correct typing error “2hyrdoxymethyl” in “ 2hydroxymethyl”.
- Line 188-189: Please specify the % of FBS used in this assay.
- Line 199: The reference 10 is repeated twice, please eliminate n. 24 changing with number 10 and reorganizing the subsequent reference list.
- Line 210: Please move Figure 1 after the respective sentences in the main text.
- Line 304. Supplementary figure S12 is not present in the supplementary file. If it's not a typing error, please upload the respective image.
- Line 304: please correct MDA-MB-231 and specify in the manuscript that is a triple negative breast cancer cell line with a more aggressive behaviour compared to other breast cancer cell lines.
- The authors tested binding ability directly on CD81 protein expressed on cell surface of MDA-MB-231 cells or other cell lines? It would be interesting to know the binding affinity/specificity of P152 peptide directly to CD81 expressed on whole cell surface by using for example flow cytometry or confocal microscopy assays. Can the authors provide additional biological informations?

Author Response
The authors would like to thank reviewer 3 for reviewing our manuscript and providing meaningful comments/suggestions. To improve our manuscript for publication in Biomolecules, we have carefully addressed your comments/suggestions in the revised manuscript as follows.
Point 1, Line 63-64:
We missed to include the reference no. 18. We have updated the referecence from “[19,20]” to “[18-20]” (Line 64).
Point 2, Line 84:
We have corrected this typing error from “2hyrdoxymethyl” to “2hydroxymethyl” (Line 84).
Point 3, Line 188-189:
We have specified the percentage (10%) of FBS used in this assay in the revised manuscript (Line 190).
Point 4, Line 199:
Thank you very much for your notice. We have eliminated reference no. 24 and updated the subsequent references. (Line 205)
Point 5, Line 210:
We have moved figure 1 after the respective sentences in the revised manuscript.
Point 6, Line 304:
We are sorry for this mistake. Actually, figure S12 was moved as Figure 5 in the main text. We have mentioned figure 5 instead of figure S12 in the revised manuscript (Line 321).
Point 7, Line 304:
We have updated the description of MDA-MB-231 cell line as follows.
Lines 297-300: Herein, MDA-MB-231 cell line was used because this is a triple negative human breast cancer cell line with a more aggressive and metastatic behavior compared to other breast cancer cell lines and shows the high expression of CD81 [14].
Point 8,
According to our previous study on a CD9-binding peptide [Ref. no. 29], the observation by high-resolution confocal microscopy suggested that the peptide directly bound to CD9 expressed on the cell surface. Thus, we hypothesize that P152 in this study has the same potential to directly bind to CD81 expressed on the cell surface. We agree with the reviewer 3’s suggestion that an additional experiment would provide strong evidence for our hypothesis. Since we do not have the accessibility to high-resolution confocal microscopy at this time, we have additionally addressed this point as our limitation and also included this point in future investigations as follows.
Lines 328-335: According to our previously-screened CD9-binding peptide, the observation by high-resolution confocal microscopy suggested that the peptide is directly bound to CD9 expressed on the cell surface [29]. Thus, we hypothesize that P152 in this study has the same potential to directly bind to CD81 expressed on the cell surface and then inhibit cancer cell migration. Due to the limited data here, the affinity, specificity, and inhibitory activity of P152 should be extensively studied by various investigations, such as co-immunoprecipitation, flow cytometry, and confocal microscopy assays, in the future.

Round 2
Reviewer 2 Report (New Reviewer)
This manuscript is a resubmission of an earlier submission. The following is a list of the peer review reports and author responses from that submission.
Round 1
Reviewer 1 Report
Dear Authors,
I have reviewed the submission by Suwatthanarak T. et al. with the title : 'Screening of EWI-2-derived peptides for targeting tetraspanin 2 CD81 and their effect on cancer cell migration'.
Several points were identified that do not allow me to recommand publication but to reject your work. Find below the most mandatory comments:
- line 57: MDA-MB-231 human breast cancer cells: is this a research cell line? metastatis are usually referred to tumors in patients. Their phenotype might be different to the cell line
- line 58-59: reconsider your sentences; compared to antibodies, peptides are susceptible to proteolytic degradation and antibody drugs are very stable (halflife of weeks).
- Sodium bicarbonate buffer inmethods: check your pH!
- protein labeling with alexa fluor 488 dye. How was the labeling performed? The dye has not reactive species. No covalent labeling is expected from this incubation of your dye with protein.
- The supplementary data were not disclosed to the reviewer!
- The rational for choosing the four BP peptides is not disclosed. From Figure 1A several spots show identical signal.
- Table 1: Are the estimated Kd values, measured data or how have these parameters been determined?
- Why have you used 100 nM for these four selected BP for the Boyden migration assay? Their Kd is much different ..... The Boyden experiment should be done in a way to allow conc.-response measurment.
- The overall significance of your study is very low. These short peptides showed in vitro activity on cancer cell migration. the reader would expect more conviencing data on the bioactivity, molecular mechanims and developmental step of the peptides toward therapeutic applications.
Author Response
Dear Reviewer 1,
We appreciate Reviewer 1 for reviewing our work and providing comments. We have carefully addressed your comments as follows.
Point 1: Line 57: MDA-MB-231 human breast cancer cells: is this a research cell line? Metastatis are usually referred to tumors in patients. Their phenotype might be different to the cell line.
Response 1: We used MDA-MB-231 human breast cancer cell line in this study because this cell line is one of the most famous cell lines, especially, for metastasis research. We have listed examples of the published research articles, of which MDA-MB-231 cell line was used for metastasis study (in vitro and in vivo) as follows.
- Vences-Catalán F, Rajapaksa R, Kuo CC, Miller CL, Lee A, Ramani VC, Jeffrey SS, Levy R, Levy S. Targeting the tetraspanin CD81 reduces cancer invasion and metastasis. Proc Natl Acad Sci U S A. 2021 Jun 15;118(24):e2018961118. doi: 10.1073/pnas.2018961118. PMID: 34099563; PMCID: PMC8214710.
- Liu D, Guo P, McCarthy C, Wang B, Tao Y, Auguste D. Peptide density targets and impedes triple negative breast cancer metastasis. Nat Commun. 2018 Jul 4;9(1):2612. doi: 10.1038/s41467-018-05035-5. PMID: 29973594; PMCID: PMC6031661.
- Hu F, Zhang Y, Li M, Zhao L, Chen J, Yang S, Zhang X. BMP-6 inhibits the metastasis of MDA-MB-231 breast cancer cells by regulating MMP-1 expression. Oncol Rep. 2016 Mar;35(3):1823-30. doi: 10.3892/or.2015.4540. Epub 2015 Dec 31. PMID: 26751737.
- Liu K, Newbury PA, Glicksberg BS, Zeng WZD, Paithankar S, Andrechek ER, Chen B. Evaluating cell lines as models for metastatic breast cancer through integrative analysis of genomic data. Nat Commun. 2019 May 15;10(1):2138. doi: 10.1038/s41467-019-10148-6. PMID: 31092827; PMCID: PMC6520398.
- Liu YL, Chou CK, Kim M, Vasisht R, Kuo YA, Ang P, Liu C, Perillo EP, Chen YA, Blocher K, Horng H, Chen YI, Nguyen DT, Yankeelov TE, Hung MC, Dunn AK, Yeh HC. Assessing metastatic potential of breast cancer cells based on EGFR dynamics. Sci Rep. 2019 Mar 4;9(1):3395. doi: 10.1038/s41598-018-37625-0. PMID: 30833579; PMCID: PMC6399327.
Point 2: line 58-59: reconsider your sentences; compared to antibodies, peptides are susceptible to proteolytic degradation and antibody drugs are very stable (halflife of weeks).
Response 2: The meaning of a sentence recommended by reviewer 1 is opposite to our sentence (Lines 58-59 in the first manuscript). However, our sentence (Lines 58-59 in the first manuscript) is consistent with a cited research article (Reference no. 16) by Liu D. et al. published in Nature Communications. Liu D. et al. stated that “The use of antibodies is hindered by size, susceptibility to degradation, and orientation of the binding epitope. In contrast, peptides exhibited strong binding affinity, induced minor immune reactivity, reduced proteolytic degradation, and increased circulation times relative to monoclonal antibodies.” (Reference no. 16). Thank you for your sentence and comment, unfortunately, we cannot apply it to the revised manuscript as it does not match our intended purpose.
Point 3: Sodium bicarbonate buffer in methods: check your pH!
Response 3: We clarified the following sentences in response to the reviewer 1’s suggestion.
“According to the user guide, 84 mg of sodium bicarbonate was first dissolved in 1 mL of MilliQ to prepare 1 M sodium bicarbonate solution. Next, CD81 solution at 0.2 mg/mL was prepared by dissolving 100 µg of CD81 in 500 µL of 0.1 M sodium bicarbonate buffer (pH 8.3).” (Lines 115-118 in the revised manuscript)
Point 4: Protein labeling with alexa fluor 488 dye. How was the labeling performed? The dye has not reactive species. No covalent labeling is expected from this incubation of your dye with protein.
Response 4: As mentioned in “2.3 Preparation of fluorescently labeled CD81 protein” section of the first manuscript (lines 113-123), Alexa Fluor™ 488 protein labeling kit (Thermo Fisher Scientific, US) was applied to fluorescently label CD81 protein in our work. This ready-to-use kit provides all materials needed to conveniently label protein with amine-reactive green fluorescent Alexa Fluor 488 dye. Notably, the used dye has the reactive group (succinimidyl ester) to label protein at primary amine under slightly alkaline pH.
To clarify this point, we updated the following sentence.
“The obtained CD81 solution was later applied into a vial containing Alexa Fluor™ 488 fluorescent dye with succinimidyl ester group, followed by incubation on a magnetic stirrer at room temperature in dark for 3 h.” (Lines 118-120 in the revised manuscript)
Point 5: The supplementary data were not disclosed to the reviewer!
Response 5: We believe that some technical problems might occur because we uploaded the supplementary file at the time of submission. We would ask the editor about it when we resubmit our revised manuscript.
Point 6: The rational for choosing the four BP peptides is not disclosed. From Figure 1A several spots show identical signal.
Response 6: We already indicated the rationale for choosing the four candidate peptides in lines 207-210 of the first manuscript, “From this result, four peptides (P97, P132, P152, and P153) with a fluorescence intensity greater than the average (AVG) and two standard deviations (SDs) were selected as candidate CD81-BPs (Figures 1A, 2 and Table 1)”.
Point 7: Table 1: Are the estimated Kd values, measured data or how have these parameters been determined?
Response 7: We described details for estimating the KD values in the section of Materials and Methods of the original manuscript (Lines 135-143). Moreover, we mentioned some details again in the Results and Discussion as follows. “The peptide arrays containing all candidates were applied in the binding assay with fluorescently labeled CD81 at different concentrations for estimating the KD value of each CD81-BP candidate (Figure S5A). Figure S5B shows the negative control (4A or AAAA peptide)-subtracted spot intensities of all candidates against CD81 concentrations. Based on the Hill equation, the KD value of P97, P132, P152, and P153 was 12.6, 1.63, 0.91, and 1.06 μM, respectively (Table 1).” (Lines 227-232 of the first manuscript)
Point 8: Why have you used 100 nM for these four selected BP for the Boyden migration assay? Their Kd is much different ..... The Boyden experiment should be done in a way to allow conc.-response measurement.
Response 8: In our earlier studies (References no. 18 and 20), we screened 8-mer CD9-targeting peptide (CD9-BP) from EWI-2 and it could significantly inhibit cancer cell migration in the Boyden assay at the concentration of 100 nM. Therefore, the peptide concentration of 100 nM was used according to our previous result (Reference no. 18).
In response to this point, the following sentence was added to the revised manuscript.
“The peptide concentration of 100 nM was chosen because a CD9-targeting peptide (CD9-BP) screened from EWI-2 in our previous study could significantly inhibit cancer cell migration at this condition [18].” (Lines 285-288 in the revised manuscript)
Point 9: The overall significance of your study is very low. These short peptides showed in vitro activity on cancer cell migration. The reader would expect more conviencing data on the bioactivity, molecular mechanims and developmental step of the peptides toward therapeutic applications.
Response 9: We respect reviewer 1’s opinion regarding the overall significance of our study. This manuscript is the first attempt to identify CD81-binding peptides (CD81-BPs) from the amino acid sequence of EWI-2 protein, a major partner of CD81, using the peptide array technique. Besides, the identified peptides showed the ability to inhibit cancer cell migration. We thus believe that our newly-screened CD81-BPs with the potential to suppress cancer cell migration are beneficial and will be of interest to the readers of this journal, Biomolecules.

Reviewer 2 Report
The authors report that peptides selected from the primary sequence of the CD9/CD81 partner protein can bind to, and affect the function of, CD81. This is very similar to a previous publication on CD9, with some of the same peptides reported. For this reason, it is of limited novelty but should still be of interest to researchers in the field.
My main concern with this manuscript is the lack of an appropriate control for the peptide. The control used is a tetramer of only Ala residues, in contrast to the octamers of mixed residues, including charged amino acids. The authors cannot say that the sequence of amino acids (derived from EWI2) is important for function, only that the overall charge of the peptide may be important. To overcome this limitation (also used in a previous publication on related tetraspanin CD9), the authors must use a scrambled sequence control peptide for at least one of the active peptides (preferably the most active), to demonstrate that the sequence of amino acids is important, rather than just the charge.
Author Response
Thank you very much for reviewing our manuscript and suggesting that our work should be of interest to researchers in the field. We have carefully addressed your comment to improve our manuscript as follows.
Point 1: My main concern with this manuscript is the lack of an appropriate control for the peptide. The control used is a tetramer of only Ala residues, in contrast to the octamers of mixed residues, including charged amino acids. The authors cannot say that the sequence of amino acids (derived from EWI2) is important for function, only that the overall charge of the peptide may be important. To overcome this limitation (also used in a previous publication on related tetraspanin CD9), the authors must use a scrambled sequence control peptide for at least one of the active peptides (preferably the most active), to demonstrate that the sequence of amino acids is important, rather than just the charge.
Response 1: Regarding the concern about the peptide overall charge, in Figure 2 and Table 1, the overall charges of the top 4 peptides were not proportional to their binding affinities or fluorescence spot intensities. P152 with a charge of +3.5 showed lower fluorescence spot intensity than P132 with the charge of +2.6 (Figure 2 and Table 1). Moreover, RRRR and KKKK control peptides with the same charge (+4) showed different spot intensities (Figure 2). These data suggest that overall charge is not only the reason for CD81 binding. We also confirmed that the affinity of EWI-2-derived peptide to tetraspanin CD9 depended on the peptide structure rather than overall charge by performing alanine scanning, sequence truncation and structural prediction in our previous publication (Reference no. 18) mentioned by reviewer 2.
In response to this comment, we have newly performed the peptide-protein docking analysis using three scrambled sequences. The sequence of our best candidate (P152, CFMKRLRK) was scrambled by an unbiased and auto sequence generator software (http://www.mimotopes.com/peptideLibrary Screening.asp?id=97) to obtain three scrambled peptides (FRKCKMRL, KKRCRFLM, RFMRCLKK). All scrambled sequences were later applied to the HPEPDOCK server for blind peptide-protein docking (Reference no. 23). Interestingly, all scrambled peptides had a higher docking energy score to CD81 than the original P152, implying the loss of CD81-binding activity in the scrambled peptides (Table S3). These data indicate that P152’s sequence was specific for CD81-binding activity rather than amino acid composition, and charge or isoelectric point, pI.
According to the comment, Table S3 was added to the supplementary file and the following sentences were added to the revised manuscript.
“However, the affinity to CD81 of candidate peptides was not proportional to the isoelectric point, charge, or grand average of hydropathy values (Figure 2 and Table 1). ” (Lines 214-216 in the revised manuscript)
“Three P152-scrambled peptides also had a higher docking energy score to CD81 than the original P152, implying the loss of CD81-binding activity in the scrambled peptides (Table S3). These data indicate that P152’s sequence was specific for CD81-binding activity rather than the amino acid composition.” (Lines 270-274 in the revised manuscript)

Round 2
Reviewer 2 Report
The authors have made an effort to accommodate my concerns about the lack of a proper control peptide. They have identified three potential control peptides with likely lower binding to the target. However, they have not synthesised one of these peptides and used it to test for actual binding and, more importantly, for function in the migration assay. I would consider the latter experiment to be essential to demonstrate the sequence specificity of the selected peptides.
Author Response
We appreciate the reviewer’s suggestion to add the scrambled peptide result. We have synthesized a scrambled sequence of P152 (P152-1, FRKCKMRL) and performed the migration assay (Boyden chamber assay). The mass spectrum and HPLC chromatogram of P152-1 were shown in Figure S10. In the Boyden chamber assay, although P152-1 could inhibit the migration of MDA-MB-231 cancer cells, P152-1 was less effective compared to P152, demonstrating the sequence-dependent activity (Figure S11). In response to this, the following description and figures were added in the revised manuscript and supplementary file.
“Especially, P152 had the highest potency to suppress MDA-MB-231 cell migration, and its scrambled peptide (P152-1, FRKCKMRL) showed less anti-migration activity (Figures 4, S10 and S11).” (pages 7-8, lines 295-297)

Round 3
Reviewer 2 Report
The authors have performed the appropriate control but the data shows that scrambled peptide and the authentic peptide have very similar effects at 100nM. Differences may be observed if the peptides are titrated but, for the present, it cannot be concluded that the CD81-derived peptides have any specific biological effect.
Author Response
Thank you for your valuable comments.
We have added another scrambled peptide data P152-2, KKRCRFLM to Figure S12. The peptide P152 only showed the statistical significance, p-value <0.05, and sequence specific activity could be shown. In response to this comment, following sentences as well as Figure S11-12 were added.
“In particular, P152 had the highest potency to suppress MDA-MB-231 cell migration (Figure 4). Compared to its scrambled peptides (P152-1 (FRKCKMRL) and P152-2 (KKRCRFLM)), only P152 origin showed the statistical significance (P-value<0.05) in the anti-migration activity (Figures S10, S11 and S12).” Page 7, line 295-page 8, line 298, section 3.2
For the dissociation constant (KD) measurement, following sentences were revised for better understanding.
“For evaluation of the dissociation constant (KD), the peptide array containing candidate CD81-BPs and a control AAAA peptide was prepared as described in section 2.2. After blocking the peptide array with 3% BSA, the peptide array was immersed in a solution of Alexa488-labeled CD81 protein (1 µM, 750 nM, 500 nM, 250 nM, 125 nM or 50 nM) at room temperature for 3 h. The peptide array was then washed with Tris-HCl buffer before scanning with Typhoon FLA 9500 imager (GE Healthcare, UK). The spot intensity of each peptide was then quantified by ImageQuant software (GE Healthcare, UK). In the final step, the KD value of each candidate peptide for the CD81 protein was determined using Origin, data analysis, and graphing software (OriginLab Corporation, US).” Page 3, line 136-144, section 2.5
